# Profiling presynaptic scaffolds using split-GFP reconstitution reveals cell-type-specific spatial configurations in the fly brain

**Hongyang Wu[1], Yoh Maekawa[2], Sayaka Eno[1], Shu Kondo[3], Nobuhiro Yamagata[1,4], Hiromu Tanimoto[1,2]***

[1]Graduate School of Life Sciences, Tohoku University, Sendai, Japan; [2]Department of Biology, Faculty of Science, Tohoku University, Sendai, Japan; [3]Department of Biological Science and Technology, Tokyo University of Science, Tokyo, Japan; [4]Department of Life Science, Graduate School of Engineering Science, Akita, Japan

## eLife Assessment

This **important** work introduces a splitGFP-based labeling tool with an analysis pipeline for the synaptic scaffold protein bruchpilot, with tests in the adult Drosophila mushroom bodies, a learning center in the Drosophila brain. The evidence supporting the conclusions is **convincing**.

*For correspondence: hiromut@m.tohoku.ac.jp

**Abstract** Characterization of intracellular synapse heterogeneity aids in understanding the intricate computational logic of neuronal circuits. Despite recent advances in connectomics, the spatial patterns of synapses and their inter-individual variability remain largely unknown. Using directed split-GFP reconstitution, we achieved visualization of endogenous Bruchpilot (Brp), a presynaptic active zone (AZ) scaffold protein, in a cell-type-specific manner. By developing a high-throughput quantification pipeline, we profiled AZ structures in identified neurons of the mushroom body circuit, where intracellular synaptic patterns are crucial due to compartmentalized connectivity. Quantitative characterization of the pattern of Brp clusters across multiple individuals revealed cell-type-dependent synaptic heterogeneity and stereotypy. Furthermore, we discovered previously unidentified sub-compartmental synapse configuration and its transient structural plasticity triggered by associative learning. These profiles reveal multilayered spatial configurations of AZs, from stereotyped overall AZ distribution patterns to local arrangements of neighboring synapses.

## Introduction

The brain is a complex network of neurons, and understanding their wiring strategies provides crucial insights into their roles in neuronal computation. Release sites in the presynaptic terminal of *Drosophila* neurons are decorated with electron-dense projections composed of active zone (AZ) scaffolds that are essential for efficient synaptic transmission (*Akbergenova et al., 2018*; *Fouquet et al., 2009*; *Kittel et al., 2006*; *Matkovic et al., 2013*; *Newman et al., 2022*; *Paul et al., 2015*; *Wagh et al., 2006*). These dense projections serve as a synapse marker, enabling the whole-brain connectomes and comprehensive mapping of synaptic connections at the electron microscopic level (*Dorkenwald et al., 2024*; *Scheffer et al., 2020*; *Schlegel et al., 2024*; *Takemura et al., 2024*; *Zheng et al., 2018*). While connectome-based approaches catalyzed the exploration of cell-type distinctions, circuit motifs, and neurotransmitters, their application to comparisons across different conditions is still challenging

due to the limited throughput of data acquisition and dense reconstruction. Since the synaptic structure and connectivity reflect substantial variability of development and experiences (*Fernández et al., 2008*; *Gilestro et al., 2009*; *Kremer et al., 2010*; *Sachse et al., 2007*; *Schlegel et al., 2024*; *Turrel et al., 2022*; *Zhang et al., 2018*), cross-individual comparisons are necessary. Fluorescence labeling offers high throughputs and contents, especially when comparing synaptic organizations across cell types (*Mosca and Luo, 2014*) or brain regions (*Gao et al., 2019*). Nevertheless, characterizing endogenous synaptic proteins in specific cells using fluorescence labeling remains challenging in the central nervous system (CNS). This has driven growing interest in designing cell-type-specific fluorescence-tagging strategies for endogenous synaptic proteins (*Chen et al., 2014*).

The circuit of *Drosophila* mushroom bodies (MBs) plays a central role in olfactory associative learning (*Davis, 2023*). The major intrinsic MB neuron, known as Kenyon cells (KCs), responds to odor stimulation and synapses onto the spatially segregated dendrites of MB output neurons (MBONs) that divide the MB into compartments. Distinct types of dopaminergic neurons (DANs) synapse onto specific compartments and thereby modulate KC-MBON synapses in the corresponding compartments (*Aso et al., 2014*). Presynaptic calcium levels in these DANs undergo sub-compartmental GABAergic modulation and inform memory specificity, postulating the distinct synapse structures at individual release sites (*Yamagata et al., 2021*). Two types of giant interneurons, the anterior paired lateral (APL) neuron and the dorsal paired medial (DPM) neuron extensively ramify across the MB lobes and provide recurrent modulations to KC synapses (*Haynes et al., 2015*; *Keene et al., 2006*; *Keene et al., 2004*; *Lin et al., 2014*; *Liu and Davis, 2009*; *Pitman et al., 2011*; *Waddell et al., 2000*; *Yu et al., 2005*). Considering their comprehensive projections throughout the MB lobes, modulation of local circuits requires intracellular tuning of synaptic structures. These studies together underscore the importance of spatially distinguishing individual synapses within a cell. However, the high AZ density in the MB lobes precluded conventional approaches of fluorescent microscopy from profiling synaptic structures of specific neurons (*Scheffer et al., 2020*).

Here, we present a spatial analysis of individual AZs within the MB circuit at the single-cell resolution, achieved by cell-type-specific visualization of the endogenous AZ protein Bruchpilot (Brp) using the CRISPR/Cas9-mediated split-GFP tagging system. *Drosophila* ELKS/CAST/ERC family member Brp plays a central role in molecular assemblies at AZs by accumulating calcium channels and synaptic vesicles (*Fouquet et al., 2009*; *Hallermann et al., 2010*; *Kittel et al., 2006*; *Matkovic et al., 2013*; *Wagh et al., 2006*). Therefore, Brp enrichment serves as a structural proxy suited for estimating synapse function, such as release probability at single AZs (*Akbergenova et al., 2018*; *Newman et al., 2022*). We developed a high-throughput quantification pipeline to systematically profile Brp clusters of individual AZs in different MB-innervating neurons. Characterizing the distinct parameters and localization of Brp clusters revealed AZ distribution stereotypy across individuals and significant synaptic heterogeneity within single neurons. These cell-type-specific synapse profiles suggest that AZs are spatially organized at multiple scales, ranging from interindividual stereotypy to neighboring synapses in the same cells.

## Results

### Establishing an experimental system for cell-type-specific profiling of AZ structures

As the molecular assembly of synapses is reported to be sensitive to the dosage of Brp, we visualized endogenous Brp instead of transgenic expression of tagged Brp (*Huang et al., 2020*). To label endogenous Brp specifically in designated cell types, we employed the split-GFP tagging system (*Kamiyama et al., 2016*; *Kondo et al., 2020*). Using CRISPR/Cas9, we inserted the $GFP_{11}$ fragment (the eleventh β-strand of the super-folder GFP) just prior to the stop codon of the *brp* gene. The $GFP_{1-10}$ fragment is expressed in cells of interest by the GAL4/UAS system. The self-assembly of the split-GFP fragments is therefore directed only in GAL4-expressing cells and visualizes endogenous Brp through reconstituted GFP fluorescence (*Figure 1A*).

As a proof of principle, we first validated the localization and plasticity of split-GFP tagged Brp (*Figure 1B*). To verify tagged Brp localization in an individual neuron, we directed GFP reconstitution while expressing a plasma membrane marker in the serotonergic DPM neuron using *VT64246-GAL4* (*Figure 1C*). Confocal microscopy revealed Brp::reconstituted GFP (Brp::rGFP) signals only in the MB

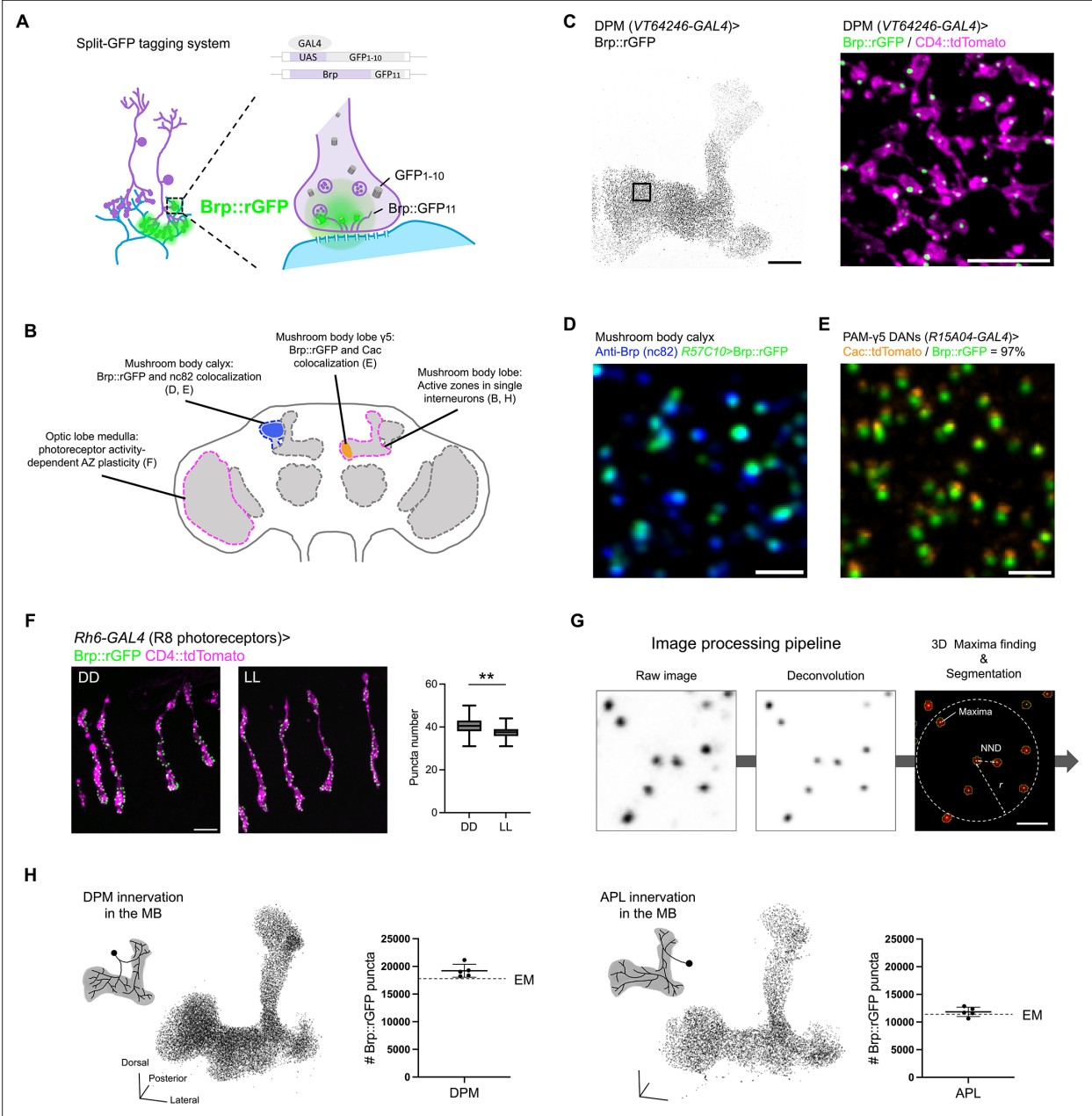

**Figure 1.** Spatial characterization of Brp::rGFP clusters at single active zone (AZ) level. (**A**) Schematic of the split-GFP tagging strategy. The $GFP_{11}$ fragment is inserted by CRISPR/Cas9 just prior to the stop codon of *brp*, while $GFP_{1-10}$ is expressed in specific cell types via GAL4-UAS system. (**B**) Brp::rGFP signals in dorsal paired medial (DPM) neurons. The entire mushroom body (MB) lobe structure is shown. The black box in the left panel indicates the zoom-in area (right panel). $GFP_{1-10}$ and CD4::tdTomato (magenta) were co-expressed using *VT64246-GAL4*. Scale bar, 20 μm (left); 5 μm (right). (**C**) Schematic showing brain regions and cell types analyzed in C–F and H. (**D**) Brp::rGFP (green) and anti-Brp immunostaining signals (blue) in the MB calyx. $GFP_{1-10}$ was expressed using pan-neuronal driver *R57C10-GAL4*. Scale bar, 1 μm. (**E**) Brp::rGFP (green) and Cac::tdTomato (orange) in PAM-γ5 dopamine neuron terminals. $GFP_{1-10}$ and Cac::tdTomato were co-expressed using *R15A04-GAL4*. Scale bar, 1 μm. (**F**) Brp::rGFP (green) in R8 photoreceptors upon constant dark or light conditions. $GFP_{1-10}$ and CD4::TdTomato (magenta) were co-expressed using *Rh6-GAL4*. Brp::rGFP cluster number of each R8 photoreceptor terminal in the medulla was quantified. n=30 (3 brain samples, 10 terminals from each brain). Welch's t-test. Scale bar, 5 μm. (**G**) Image processing pipeline. Raw images were processed with image deconvolution to improve the signal quality. 3D maxima (green pixels) are detected for each Brp::rGFP cluster. 3D regions of interest (ROIs) are indicated by yellow circles. Nearest neighbor distance (NND), *r* is defined as 3× mean NND. Scale bar, 1 μm. (**H**) Number of Brp::rGFP clusters detected in DPM and anterior paired lateral (APL) neurons. The dashed lines indicate the number of electron-dense projections identified in the hemibrain connectome for DPM (17530) and APL (10789) on the MB lobes and peduncle (*Scheffer et al., 2020*). Schematics illustrate the innervation patterns of DPM and APL neurons. For both DPM and APL, n=5. Lower panels show the representative 3D reconstructions of Brp::rGFP clusters in DPM and APL. Scale bars, 20 μm. Data were presented as mean ± SEM.

*Figure 1 continued on next page*

*Figure 1 continued*

The online version of this article includes the following figure supplement(s) for figure 1:

**Figure supplement 1.** Split-GFP tagging does not affect Bruchpilot (Brp) neuronal expression specificity and expression pattern in the brain.

**Figure supplement 2.** Neuron-specific assembly of Brp::rGFP.

**Figure supplement 3.** Rab3 knockdown in Kenyon cells (KCs) increases Brp::rGFP intensity.

**Figure supplement 4.** Effect of iterative Richardson-Lucy image deconvolution.

**Figure supplement 5.** Heatmaps of F-scores showing the performance of the pipeline in detecting individual Brp::rGFP clusters in different cell types.

lobes, aligning with DPM terminals (*Waddell et al., 2000*). These signals were distributed as discrete clusters within plasma membrane varicosities resembling presynaptic boutons (*Figure 1C*), suggesting that Brp::rGFP clusters labeled individual AZs in the DPM neuron. AZ localization of Brp::rGFP was further validated by co-localization with anti-Brp immunostaining in the calyx and the voltage-gated calcium channel subunit Cacophony (Cac) in the PAM DANs (*Figure 1D and E*; *Kawasaki et al., 2004*; *Kittel et al., 2006*). We found that GFP$_{11}$ tagging did not affect *brp* mRNA levels, but slightly decreased protein expression without altering overall expression pattern in the brain (*Figure 1—figure supplements 1–2*). Brp::rGFP successfully visualized previously reported light-induced AZ plasticity in the photoreceptors (*Sugie et al., 2015*). Constant light exposure reduced the numbers of Brp::rGFP clusters in the R8 terminals in the medulla, compared to constant darkness (DD) (*Figure 1F*). This was further substantiated by increased Brp::rGFP cluster intensities upon knocking down *rab3* expression (*Figure 1—figure supplement 3*), recapitulating Rab3-dependent regulation of Brp allocation in the presynaptic terminals of the motor neurons (*Graf et al., 2009*).

To systematically profile Brp::rGFP clusters, we developed an image processing pipeline (*Figure 1G*). In brief, image deconvolution was applied to reduce out-of-focus light and enhance signal sharpness (*Figure 1—figure supplement 4*). Individual Brp::rGFP clusters were then segmented using the 3D Suite plugin in Fiji (*Ollion et al., 2013*), generating the 3D regions of interest (3D ROIs) that encircle each cluster. Brp::rGFP signal intensity, volume, and location of individual clusters were measured using the 3D ROIs. To optimize the segmentation, we systematically varied the parameters and calculated the F-score to measure detection accuracy against manually defined ground truth (see Materials and methods). The F-score is near 1.0 in segmenting Brp::rGFP clusters within DPM and APL neurons and achieved >0.90 in KCs with certain combinations of segmentation parameters (*Figure 1—figure supplement 5*).

To confirm that Brp clusters represent AZs, we quantified the numbers of Brp::rGFP clusters in DPM and APL neurons specifically and compared them to the connectome data. We found that cluster counts closely matched the number of AZs annotated by the hemibrain connectome (*Figure 1C*; *Scheffer et al., 2020*). This result validated our method that accurately isolated Brp::rGFP clusters corresponding to single AZs.

## AZ profiles within KCs are compartmentally heterogeneous

Brp accumulation at individual AZs is known to be heterogeneous even within a single motor neuron (*Akbergenova et al., 2018*; *Ehmann et al., 2014*; *Gratz et al., 2019*; *Paul et al., 2015*). To study such AZ heterogeneity in CNS neurons, we quantified the signal intensity of individual Brp::rGFP clusters in MB lobes. Using KC subtype-specific drivers to direct split-GFP reconstitution in γ KCs (*MB009B-GAL4*), α'/β' KCs (*MB370B-GAL4*), and α/β KCs (*MB008B-GAL4*), we found that within-cluster Brp::rGFP intensities are highly diverse in all KC subtypes (*Figure 2A*).

To visualize spatial patterns of Brp::rGFP intensity in each KC subtype, we reconstructed 3D distribution of Brp::rGFP clusters and color-coded them by signal intensities (*Figure 2B*). We found that clusters showed noticeable intensity differences between compartments in all subtypes. Even adjacent compartments, such as γ1 and γ2, β'2 and β'1, α2 and α3, showed drastic differences (*Figure 2B*). Importantly, these compartmental patterns were stable across individuals (*Figure 2C*; *Wu et al., 2025*). This result suggests that AZs in different compartments have distinct structures. Since individual KCs of each subtype synapse onto all compartments (*Scheffer et al., 2020*; *Schlegel et al., 2023*; *Takemura et al., 2024*; *Zheng et al., 2018*), this Brp compartmental heterogeneity is likely formed intracellularly.

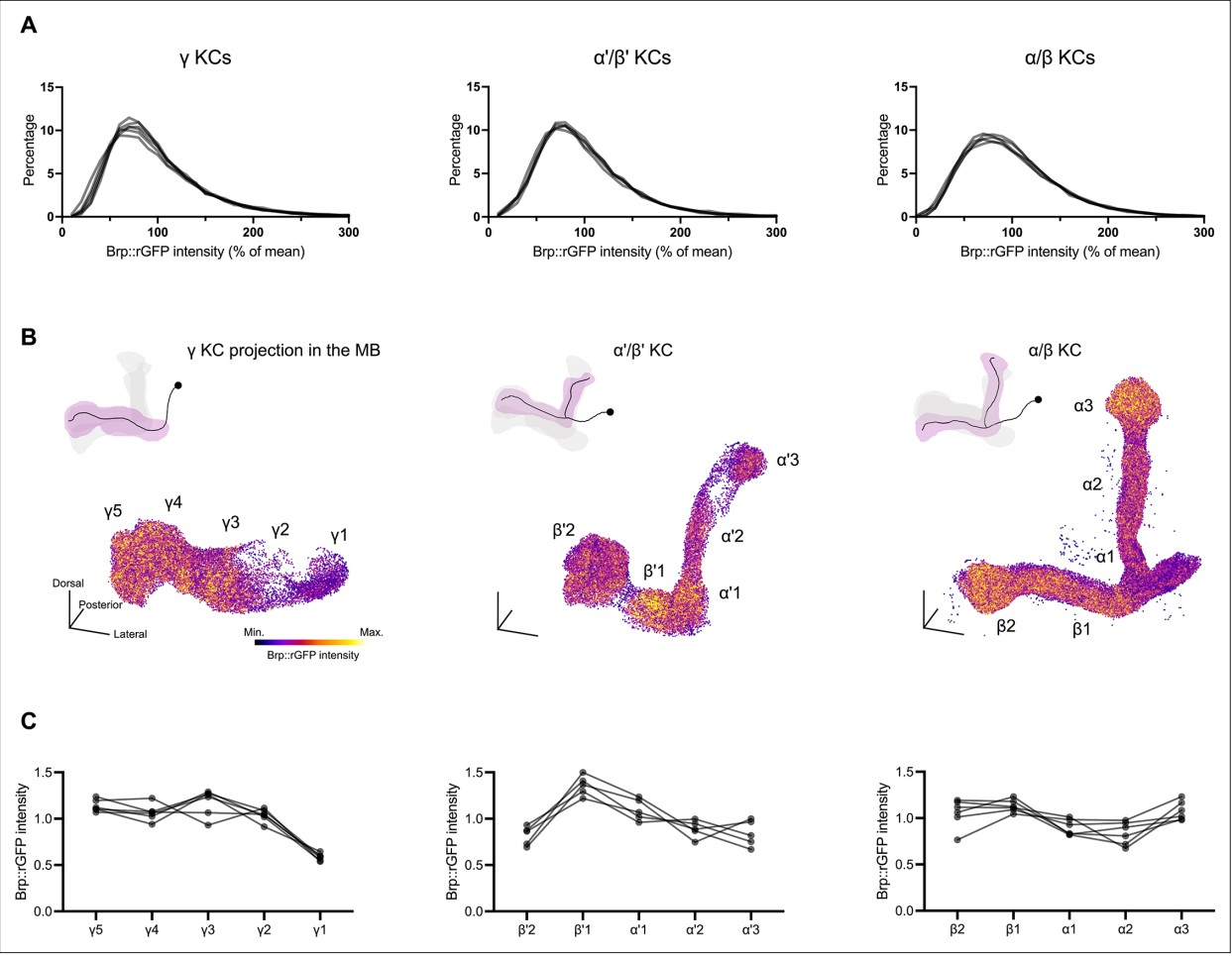

**Figure 2.** Compartmentalized active zone (AZ) structures of Kenyon cells (KCs). (**A**) Histograms of the signal intensity of individual Brp::rGFP clusters from three KC subtypes. Values are normalized by the mean intensity in each dataset (each sample). Each line represents the histogram of one independent sample. γ KCs (*MB009B-GAL4*), n=6; α/β KCs (*MB370B-GAL4*), n=5; α'/β' KCs (*MB008B-GAL4*), n=5. (**B**) 3D reconstructions of Brp::rGFP clusters in three KC subtypes, colored by the Brp::rGFP intensity. The approximate locations of compartments are as indicated. Schematics illustrate the innervation patterns of different KC subtypes. Min. and Max. were set to represent the lowest and highest 5% of Brp::rGFP intensity value in the dataset, respectively. Scale bars, 20 μm. (**C**) Signal intensity of Brp::rGFP clusters in each compartment. Brp::rGFP clusters were quantified compartmentally. Medians in different compartments are shown as the ratio against the average of five compartments in the corresponding KC subtype. Each line represents one independent sample.

## Intracellular heterogeneity of Brp concentration at individual AZs and cell-type-dependent diversity

Super-resolution microscopy has shown that the Brp molecular density at individual AZs is dynamically adjusted in motor neurons (*Ghelani et al., 2023*; *Mrestani et al., 2021*). We hypothesized that the amount of Brp does not necessarily correlate with AZ cluster size. To verify this, we calculated the Brp concentration by quantifying both the volume (number of voxels within a 3D ROI) and the Brp::rGFP intensity of each cluster. We found that clusters with similar sizes can exhibit vastly different signal intensities in the DPM neuron, suggesting the heterogeneity of Brp concentration at individual AZs (*Figure 3A*). Correlation analysis between the Brp::rGFP intensity and cluster volume revealed a substantial variability among cell types (*Figure 3B–E*). There were even significant differences among KC subtypes. These results suggest cell-type-dependent variability of the Brp concentration of individual clusters.

## Stereotypy of intracellular distribution of AZs

To study the spatial distribution of Brp clusters within single neurons, we focused on DANs. *MB504B-GAL4* labels four types of single PPL1 DANs projecting to distinct compartments (*Vogt*

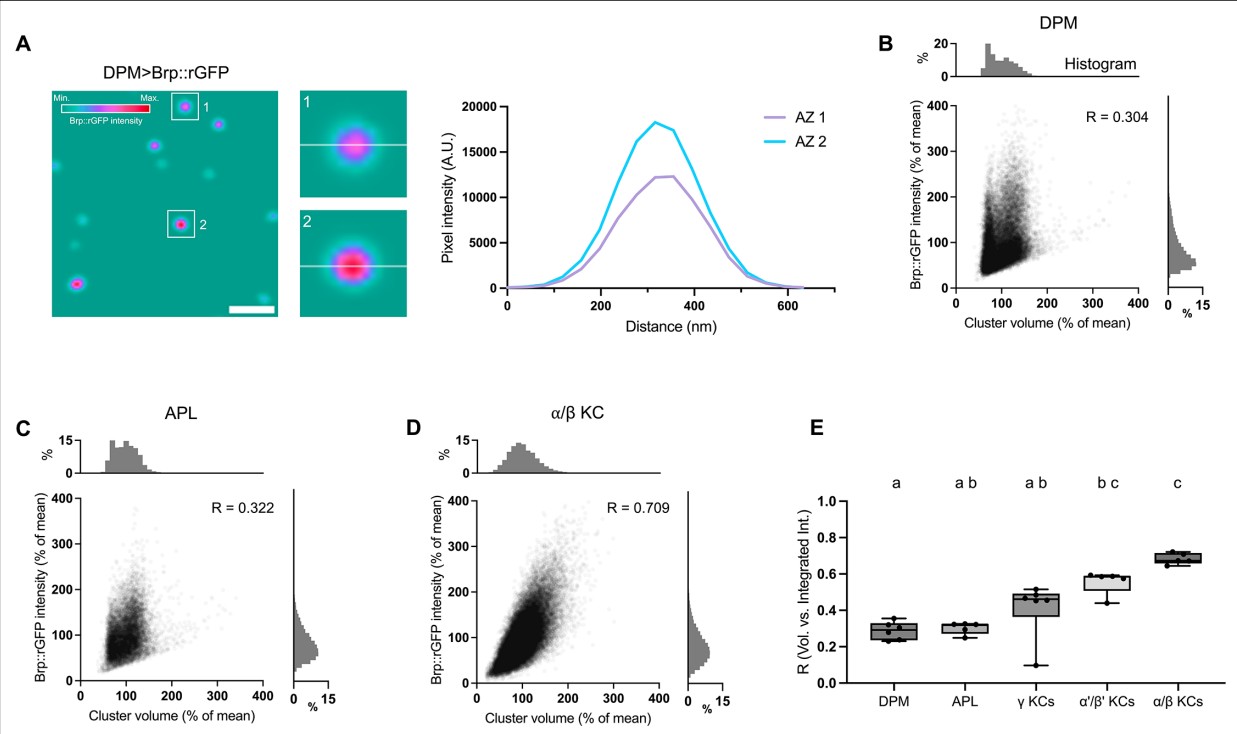

**Figure 3.** The variability of Brp::rGFP concentrations depends on cell type. (**A**) Brp::rGFP clusters with distinct intensities but similar size in dorsal paired medial (DPM) neuron. The left panel shows a cropped image of Brp::rGFP in DPM. White boxes indicate areas zoomed in for panels 1 and 2. Transparent white lines in panels 1 and 2 show the lines on which the intensity profiles were plotted. Intensity profiles were plotted for AZ1 (magenta) and AZ2 (blue), respectively. (**B–D**) Scatter plots showing the correlation between the Brp::rGFP intensity and cluster volume in DPM neuron, anterior paired lateral (APL) neuron, and α/β Kenyon cells (KCs). Data from three representative samples were shown. Pearson's correlation coefficient R was calculated for each sample. (**E**) Correlations between the intensity and volume of Brp::rGFP clusters in different cell types. DPM (n=6), APL (n=5), γ KCs (n=6, *MB009B-GAL4*), α'/β' KCs (n=5, *MB370B-GAL4*), α/β KCs (n=5, *MB008B-GAL4*). DPM vs. α'/β' KCs: p=0.0253; DPM vs. α/β KCs: p=0.0009; APL vs. α/β KCs: p=0.0035; γ KCs vs. α/β KCs: p=0.0418. Values marked with different lowercase letters represent significant difference (p<0.05). Data were presented as box plots showing center (median) and whiskers (Min. to Max.).

et al., 2014). Using this driver, we analyzed Brp::rGFP distribution in each compartment and found it uneven in terminals of both PPL1-α2α'2 and α3 (*Figure 4A and B*). To better characterize cluster distribution, we calculated the 'AZ density' defined as the number of Brp::rGFP clusters surrounding a particular cluster within a specified radius (see *Figure 1A* and Materials and methods for details). By color-coding Brp::rGFP clusters according to their AZ density, we found that AZs in PPL1-α3 were more localized to the core of the α3 compartment. This pattern was consistent across individuals (*Figure 4C and D*), suggesting a stereotyped AZ distribution within PPL1-α3 neurons.

We extended the analysis to two other single pairs of neurons innervating the MBs, the DPM and APL neurons. Visualizing AZ density revealed a stereotyped and compartment-specific distribution in the DPM neuron (*Figure 5A*). Specifically, the AZ density was constantly high in the α'/β' lobes across all individuals, supporting the known functional importance of DPM branches in the α'/β' lobes (*Keene et al., 2006*). In contrast, this compartmental pattern was less pronounced in the APL neuron. Notably, the ratio of AZ density between α3 and α'3 is strikingly different among individuals (*Figure 5B*). Taken together, these data suggest that the individual variability in AZ distribution depends on the cell type.

## Regulation of spatial configuration of AZs
Characteristics of Brp clusters and AZ distribution in KCs revealed spatial organization at the level of compartments. Recent studies further showed intra-compartmental variability of synaptic plasticity upon associative learning, prompting us to analyze sub-compartmental organization of AZs (*Bilz et al., 2020*; *Davidson et al., 2023*). We thus examined potential local patterns among AZs in proximity by calculating the correlation between nearest neighbor clusters in terms of their Brp::rGFP

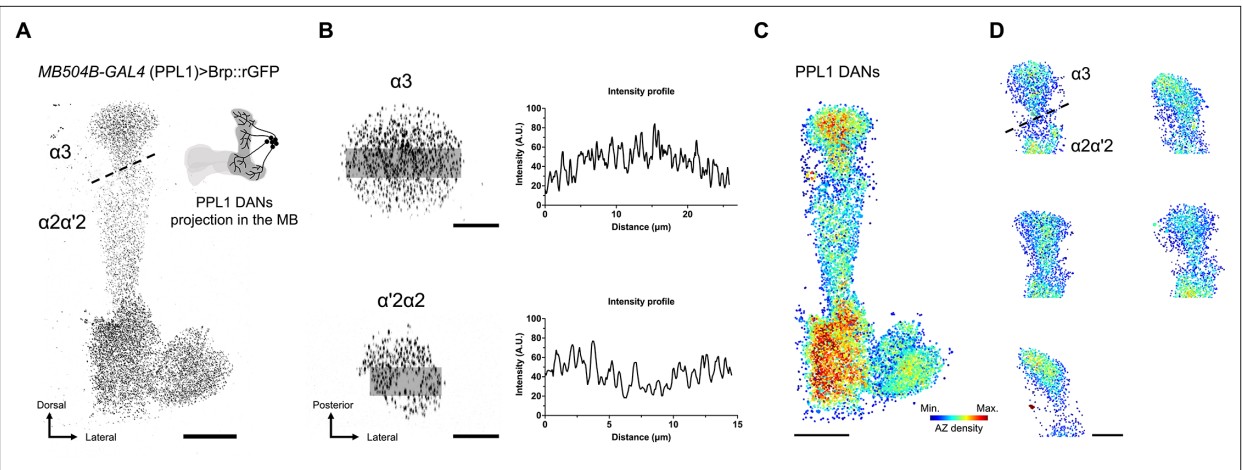

**Figure 4.** Stereotyped active zone (AZ) distribution of PPL1-α3 dopaminergic neuron (DAN). (**A**) Brp::rGFP in PPL1 DANs. UAS-GFP$_{1-10}$ was expressed using *MB504B-GAL4*. Dashed line marks the rough boundary between α3 and α2α'2 compartments. Schematic illustrates the innervation patterns of PPL1 DANs within the mushroom body (MB). Scale bar, 20 μm. (**B**) Brp::rGFP intensity profiles of α3 and α2α'2 compartments. Left panels show the max-projection images of α3 and α2α'2 optical coronal sections. Transparent gray stripes indicate areas where intensity profiles are plotted in the right panels. Scale bars, 10 μm. (**C**) 3D reconstruction colored by the AZ density in PPL-1 DANs. Color scale: Min.=0, Max.=40. Dashed line indicates the rough boundary between α3 and α2α'2 compartments. Scale bar, 20 μm. (**D**) Stereotyped AZ distribution pattern in PPL1-α3 across individuals. 3D reconstructions show the AZ density across different brain samples. Scale bar, 20 μm.

intensities (*Figure 6A and B*). This analysis revealed strong correlations in KCs and APL but not in DPM (*Figure 6C*).

We further analyzed the sub-compartmental AZ patterns by examining the spatial distribution of Brp::rGFP intensity. To reduce the high-frequency variability of Brp::rGFP intensities (*Figure 2B*), we

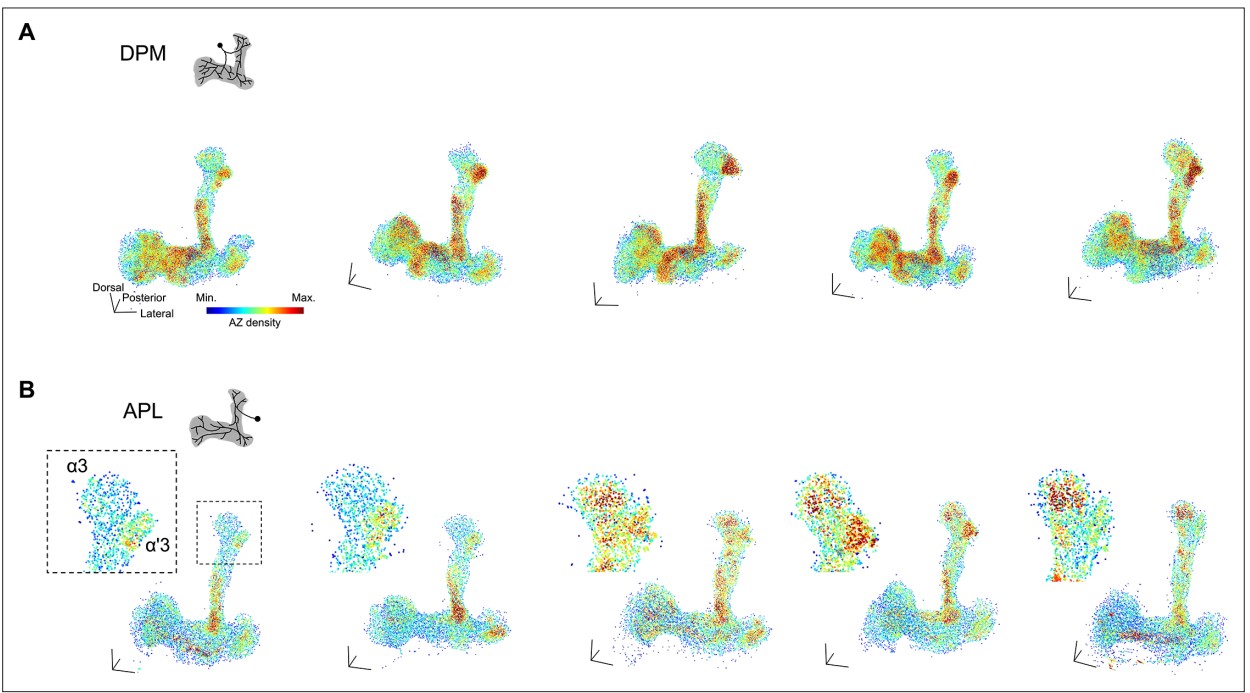

**Figure 5.** Cell-type-specific stereotypy of active zone (AZ) spatial distributions. (**A**) 3D reconstructions of Brp::rGFP clusters in dorsal paired medial (DPM) neurons, colored by AZ density. Color scale: Min.=0, Max.=35. Black arrows indicate consistently high AZ density regions across brain samples. Scale bars, 20 μm. (**B**) 3D reconstructions of Brp::rGFP clusters in anterior paired lateral (APL) neurons, colored by AZ density. Color scale: Min.=0, Max.=30. Black dashed square indicates the area zoomed in. APL reconstructions are arranged from left to right according to the overall AZ density in α3.

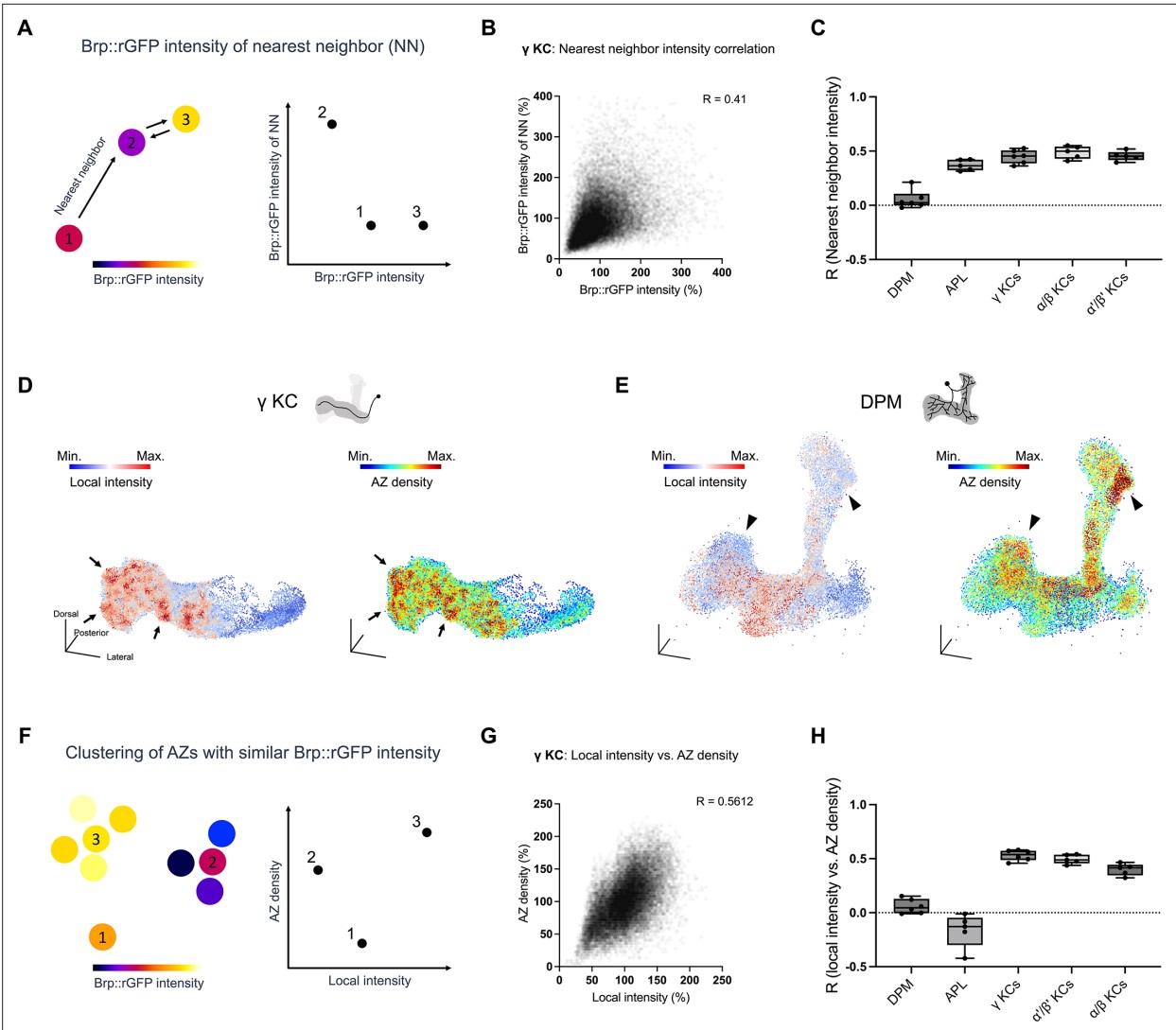

**Figure 6.** Local intensity analysis revealed sub-compartmental active zone (AZ) structures. (**A**) Brp::rGFP correlation analysis between nearest neighbors (NN). The intensity of a cluster is plotted on the x-axis, and the intensity of its nearest neighbor is plotted on the y-axis. High correlation indicates nearest neighbors have similar Brp::rGFP intensity. (**B**) Scatter plots showing the Brp::rGFP intensity correlation between nearest neighbors in a representative sample of γ Kenyon cells (KCs). Values were normalized to the mean. Pearson's correlation coefficient R is shown. (**C**) Correlation of Brp::rGFP intensities between nearest neighbor AZs in different cell types. Brp::rGFP intensities were log transformed. Dorsal paired medial (DPM) neurons vs. γ KCs: p=0.0066; DPM vs. α'/β' KCs: p=0.0027; DPM vs. α/β KCs: p=0.0066; Scale bar: 1 μm. Data were presented as box plots showing center (median) and whiskers (Min. to Max.). (**D**) 3D reconstructions of Brp::rGFP clusters in a γ KC sample, colored by local intensity and AZ density. Black arrows indicate areas with both high local intensities and high AZ densities. Min. and Max. were set to represent the lowest and highest 5% in local intensity value, respectively. Color scale of AZ density: Min.=0 and Max = 35. (**E**) 3D reconstructions of Brp::rGFP clusters in a DPM neuron, colored by local intensity and AZ density. Black triangle arrows indicate areas with high AZ densities but low local intensities. Min. and Max. are set to represent the lowest and highest 5% in local intensity value, respectively. Color scale of AZ density: Min.=0, Max.=35. (**F**) Correlation analysis between local intensity and AZ density. The local intensity of a cluster is plotted on the x-axis and its AZ density is plotted on the y-axis. (**G**) Scatter plots showing the correlation between Brp::rGFP local intensity and AZ density in a representative sample of γ KCs. Values were normalized to the mean. Pearson's correlation coefficient R is shown. (**H**) Correlations between AZ density and local intensity in different cell types. DPM (n=6), anterior paired lateral (APL) (n=5), γ KCs (n=6, *MB009B-GAL4*), α'/β' KCs (n=5, *MB370B-GAL4*), α/β KCs (n=5, *MB008B-GAL4*). DPM vs. γ KCs: p=0.0171; DPM vs. α'/β' KCs: p=0.0171; APL vs. γ KCs: p=0.0014; APL vs. α'/β' KCs: p=0.0014; APL vs. α/β KCs: p=0.0171. Data were presented as box plots showing center (median) and whiskers (Min. to Max.)

calculated the 'local intensity' for each cluster by applying mean filtering (the average Brp::rGFP intensity of all surrounding clusters within a specified radius *r*, see Materials and methods for details). By visualizing the local intensity in γ KCs, we identified AZ 'hot spots', sub-compartment-sized small groups of AZs with high local intensities, on top of the compartmental differences (*Figure 6D*). Since

Brp molecules are clustered more in AZ-dense boutons of a single motor neuron (*Paul et al., 2015*), we hypothesized that these hot spots correspond to regions with higher AZ densities. Indeed, we found that hot spots appear in high AZ density regions in γ KCs (*Figure 6D*). Consistently, the correlation between the local intensity and the AZ density was high in all KC subtypes. In contrast, for DPM and APL neurons, the local intensity of Brp::rGFP was not associated with AZ density (*Figure 6E–H*). Taken together, these results suggest previously unidentified sub-compartmental synaptic configuration organized across individual KCs.

## Associative learning re-organizes sub-compartmental synaptic configuration

Sub-compartmental synaptic configuration may undergo experience-dependent changes, such as through associative learning (*Turrel et al., 2022*; *Zhang et al., 2018*). To examine such structural plasticity in local AZ configurations (*Figure 6D and F–H*), we visualized Brp::rGFP specifically in KCs using *R13F02-GAL4* (*Figure 7A*). This induction did not alter MB morphology and short-term memory compared to wild-type (*Figure 7—figure supplements 1–2*). We presented flies with odor (4-methylcyclohexanol) with a concomitant (paired) or shifted (unpaired) electric shock (*Figure 7B*). Since both paired and unpaired groups were exposed to the same odor and electric shock, the difference between these groups purely represents the effect of association. As single-odor conditioning induced memory that decayed gradually over 1 day (*Figure 7B*), we measured the correlation between the AZ density and local intensity of individual Brp::rGFP clusters (*Figure 6F–H*) at different time points after conditioning (from 3 min to 1 day). Strikingly, we found that associative learning induced a transient modification to the KC synaptic configurations in a compartment-specific manner (*Figure 7C*). This learning-induced structural plasticity was specific at 90 min after conditioning and disappeared within 1 day (*Figure 7D*), consistent with previous studies reporting short-lived AZ remodeling in the MBs by conditioning (*Turrel et al., 2022*; *Zhang et al., 2018*). These results showcase the advantage of the split-GFP system in reporting the dynamics of local synapse subsets with high throughputs and contents. Altogether, we suggest that associative learning induces transient memory traces of local structural plasticity in AZ configuration.

## Discussion

By leveraging the CRISPR/Cas9 genome editing of the *brp* locus and split-GFP technique, we systematically profiled presynaptic structures of MB neurons in a cell-type-specific manner. This approach enabled high-throughput analysis of multiple brain samples and detailed characterizations, revealing previously inaccessible structural features of AZs in CNS neurons. Cell-type-specific labeling of other endogenous synaptic proteins, such as vesicle-associated proteins, may provide complementary insights into AZ profiling and allow molecular reconstruction of synapses.

Single-cell level analysis of AZ profiles revealed cell-type-specific stereotypy in spatial regulation (*Figures 4–6*). We found that AZ density in the DPM neurons is consistently high in the α'/β' lobes across individuals (*Figure 5*). This AZ stereotypy of DPM neurons may explain the functional significance of the branches in the α'/β' lobes. Overexpression of a Down syndrome cell adhesion molecule variant in the DPM neurons was shown to disrupt their innervation except in the α'/β' lobe, while memory formation remained unaffected (*Keene et al., 2006*). Similarly, our analysis identified the intracellular stereotypy in AZ distribution of the PPL1-α3 DAN within the α3 compartment (*Figure 4*). This sub-compartmental organization of the single DAN and layered projections of α/β KCs may underlie the differential dopaminergic modulation of KC subtypes in associative learning (*Perisse et al., 2013*). While such stereotypy is less pronounced in APL, we found strong correlations between Brp::rGFP intensities of neighboring AZs in contrast to DPM (*Figure 6D*). This suggests that AZs in APL neurons are organized more locally, allowing them to modulate microcircuits composed of KCs, DANs, and MBONs (*Amin et al., 2020*). Collectively, the spatial scale of AZ organizations depends on cell types, reflecting the function of individual MB neurons.

We observed that Brp::rGFP intensity correlates strongly with the AZ size in KC subtypes, in contrast to DPM and APL neurons (*Figure 3*). Previous studies using localization microscopy showed that the molecular density of synaptic proteins at AZs can be dynamically adjusted in motor neurons, potentially modulating the synaptic vesicle release (*Ghelani et al., 2023*; *Mrestani et al., 2021*;

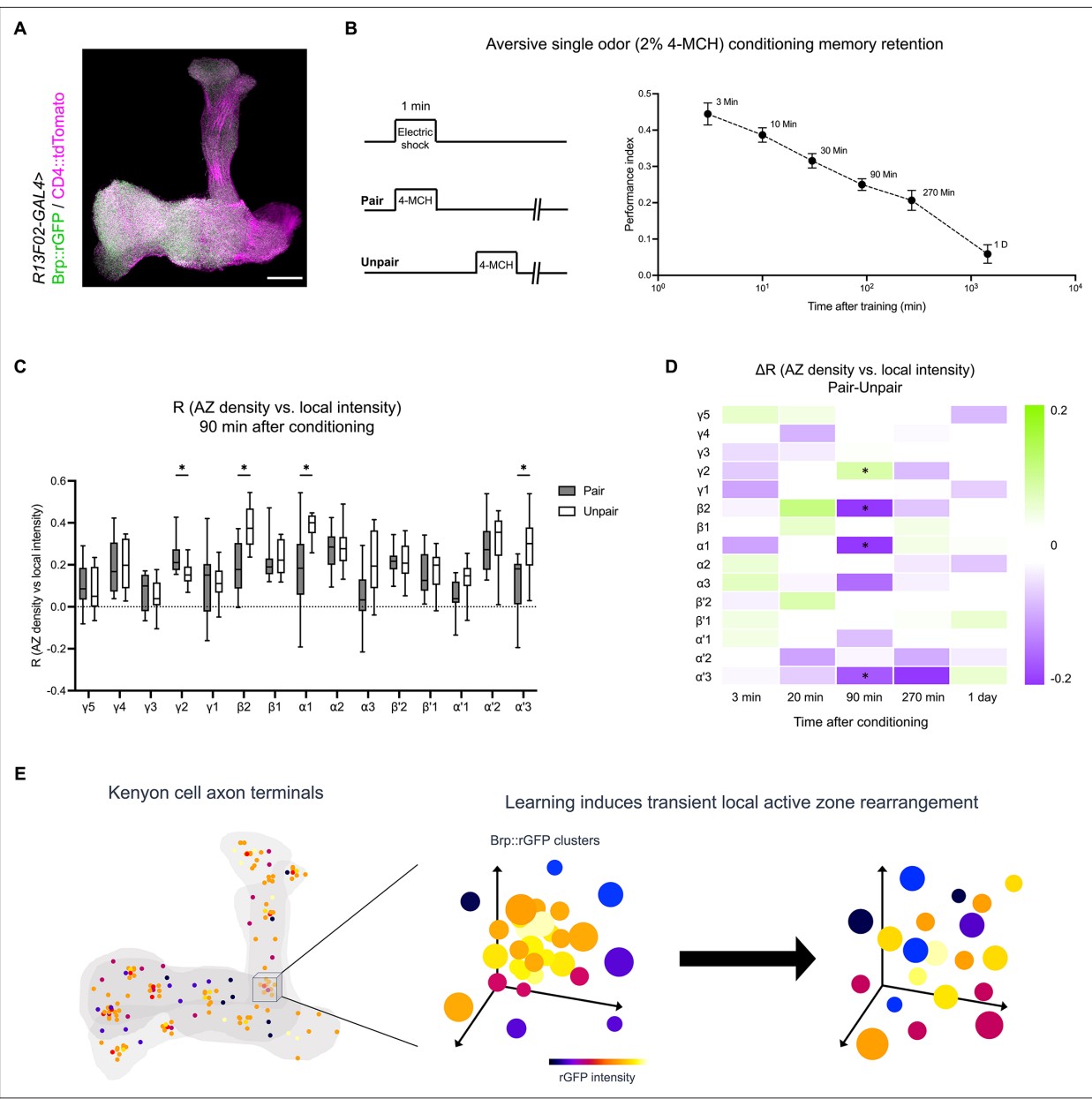

**Figure 7.** Associative conditioning re-organizes sub-compartmental active zone (AZ) clusters. (**A**) Brp::rGFP (green) and CD4::tdTomato plasma membrane marker (magenta) in Kenyon cells (KCs), visualized using *R13F02-GAL4*. (**B**) Single-odor conditioning induces long-lasting memory. Left panel, the experimental design of the aversive single-odor conditioning. The paired group received a concurrent presentation of 2% 4-MCH and 90 V electric shock. The unpaired group first receives the electric shock and then 4-MCH 1 min later. Right panel, preference of *Canton S* between 2% 4-MCH and paraffin oil at 3, 10, 30, 90, 270 min, and 1 day after single-odor conditioning. n=8 for all groups. Error bars show SEM. (**C**) Correlation coefficient (AZ density vs. local intensity of individual Brp::rGFP clusters) of each compartment at 90 min after conditioning. Pair (n=11) vs. Unpair (n=12), γ2 (p=0.0261), β2 (p=0.0060), α1 (p=0.0026), α'3 (p=0.0026). (**D**) Heatmap showing the difference of correlation coefficient (AZ density vs. local intensity of Brp::rGFP clusters) between the pair and unpair group at different time points after conditioning. The color indicates the difference, and the asterisks in the compartments indicate the significant difference. For 3 min, Pair (n=12), Unpair (n=11); For 20 min, Pair (n=12), Unpair (n=11); For 90 min, Pair (n=11) vs. Unpair (n=12), p=0.0247; For 270 min, Pair (n=11), Unpair (n=11); For 1 D, Pair (n=12), Unpair (n=9); For all results involved statistical comparison, only significant results are shown. Scale bars, 20 μm. Mann-Whitney test with original false discovery rate method of Benjamini and Hochberg correction. *p<0.05. Data were presented as box plots showing center (median) and whiskers (Min. to Max.). (**E**) Schematics showing learning-induced local AZ remodeling. Single-odor aversive training transiently dismisses high Brp level – high AZ density hot spots in specific compartments of KC terminals.

The online version of this article includes the following figure supplement(s) for figure 7:

**Figure supplement 1.** Gross anatomy of Kenyon cell (KC) terminals is not affected by Brp::rGFP tagging.

**Figure supplement 2.** Brp::rGFP tagging in Kenyon cells (KCs) does not affect short-term memory performance.

*Sachidanandan et al., 2023*). The variable Brp concentrations (i.e. the ratio of the Brp::rGFP intensity to the AZ size) in DPM and APL neurons may represent AZs with distinct Brp densities and therefore different functional modes (*Figure 3*; *Mrestani et al., 2021*). Notably, a recent study successfully predicted the neurotransmitter type of neurons using the ultrastructure of dense projections as one of the parameters (*Eckstein et al., 2024*), suggesting neurotransmitter-specific AZ structures. Interestingly, both the DPM and APL neurons are reported to release multiple neurotransmitters (*Davie et al., 2018*; *Haynes et al., 2015*; *Lee et al., 2011*; *Liu and Davis, 2009*; *Wu et al., 2013*; *Zeng et al., 2023*). Distinct Brp concentrations at AZs might thus support particular release modes or neurotransmitter types.

In KCs, the spatial analysis of local AZ configurations revealed strong correlations between Brp::rGFP intensities at neighboring AZs, as well as AZ 'hot spots', regions with high Brp localization and AZ densities (*Figure 6*). These results suggest previously unidentified, highly localized sub-compartmental AZ structures, which are likely organized across individual KCs. Presynaptic terminals of a single KC are found to be highly heterogeneous in terms of their plasticity upon dopamine modulation and associative learning (*Bilz et al., 2020*; *Davidson et al., 2023*). A recent study found that the sub-compartmentally heterogeneous activities of dopamine terminals explain memory specificity (*Yamagata et al., 2021*). Such locally distinct dopamine input may be a source of the AZ hot spot formation in KC terminals. This model is consistent with the localization of two opposing dopamine receptors Dop1R1 and Dop2R to AZs (*Hiramatsu et al., 2024*). Furthermore, we showed that the local configuration undergoes structural plasticity upon associative learning that involves presynaptic dopaminergic modulation (*Aso et al., 2012*; *Aso et al., 2010*; *Aso and Rubin, 2016*; *Burke et al., 2012*; *Liu et al., 2012*; *Yamagata et al., 2015*; *Figure 7*). Since cAMP signaling plays a crucial role in the AZ structural plasticity, especially during associative memory formation (*Baltruschat et al., 2021*; *Sachidanandan et al., 2023*; *Wu et al., 2025*), it may underlie learning-induced plasticity in sub-compartmental AZ configuration (*Turrel et al., 2022*; *Zhang et al., 2018*).

## Materials and methods
### Animals
Flies were maintained on standard cornmeal food at 25°C under a 12:12 hr light-dark cycle for all experiments. All flies used for experiments are adult males aged 3–7 days. Flies were transferred to fresh food vials after hatching and flipped every 2 days before experiments. The GAL4-UAS system was used to express transgenes, and balancers were removed for animals used in all experiments. Fly strains and resources used in this study are as follows: *MB008B-GAL4* (BDSC 68291) (*Aso et al., 2014*), *MB009B-GAL4* (BDSC 68292) (*Aso et al., 2014*), *MB370B-GAL4* (BDSC 68319) (*Aso et al., 2014*), *MB504B-GAL4* (BDSC 68329) (*Aso et al., 2014*), *VT64246-GAL4* (VDRC 204311) (*Tirian and Dickson, 2017*), *GH146-GAL4* (BDSC 30026) (*Stocker et al., 1997*), *R58E02-GAL4* (BDSC 41347) (*Pfeiffer et al., 2008*), *R13F02-GAL4* (BDSC 48571) (*Pfeiffer et al., 2008*), *R86E01-GAL4* (BDSC 45914) (*Jenett et al., 2012*), *Amon-GAL4* (BDSC 30554) (*Rhea et al., 2010*), *Rh6-GAL4* (BDSC 7464, by Dr. Claude Desplan), *UAS-CD4::tdTomato* (BDSC 35841, by Dr. Yhu Nung Jan & Chun Han), *UAS-Rab3-RNAi* (BDSC 34655) (*Perkins et al., 2015*), *brp::GFP$_{11}$* (this study and *Wu et al., 2025*), *UAS-GFP$_{1-10}$* (*Kondo et al., 2020*).

### Generation of *brp::GFP$_{11}$*
CRISPR/Cas9-mediated homologous recombination was used to insert GFP$_{11}$ right before the stop codon of *brp*. The donor vector contained a left homology arm of 747 bp, the GFP11 sequence (*Kondo et al., 2020*), a floxed 3xP3-RFP marker, and a right homology arm of 741 bp. A guide RNA sequence (TCGCAAAGCCACAGATACAC) targeting the *brp* locus was cloned into pBFv-U6.2 (*Kondo and Ueda, 2013*) to generate a gRNA expression vector. The donor and gRNA vectors were co-injected into fertilized eggs carrying the nos-Cas9 transgene (*Kondo and Ueda, 2013*). Successful transformants were identified by eye-specific RFP fluorescence. Flies were then crossed to a CyO-Cre balancer to remove the RFP marker. Primers used to clone homology arms are as follows (lowercase letters indicate homologous recombination regions with the donor vector): left arm forward: gctt gatatcgaattcAAGGACATCGAGGAAAAGGAGAAGAAG; left arm reverse: agttgggggcgtaggGAAAA AGCTCTTCAAGAAGCCAGCTGGTCC; right arm forward: tagtataggaacttcTCGCAAAGCCACAGATA

CACACATCTTGG; right arm reverse: cgggctgcaggaattcATCGTCGATAATTGTAAGTGCATGCTG. As the C-terminal of Brp is critical for molecular interaction (*Hallermann et al., 2010*), C-terminal tagging of Brp is not homozygous viable (*Kohl et al., 2014*; *Wagh et al., 2006*), potentially interfering with the interaction. All experiments in this study thus employed heterozygous animals for *brp::GFP$_{11}$*.

## qRT-PCR experiment

Procedures are as previously described (*Saito et al., 2025*). For each sample, RNA extract from ~20 heads was purified using TRIzol LS Reagent (Invitrogen) and Direct-zol RNA Microprep kit (Zymo Research), following the provided protocol. The concentration of extracted RNA samples was then measured using the NanoDrop Spectrophotometer (Thermo Fisher Scientific) and adjusted to a similar level using diluted water. Reverse transcription was performed using the ReverTra Ace qPCR RT Master Mix (TOYOBO, FSQ-301). qRT-PCR was performed using iTaq Universal SYBR Green Supermix (Bio-Rad) and CFX96 Touch Real-Time PCR Detection System (Bio-Rad). 'No template controls' were also prepared and measured alongside. Three primer pairs for *brp* were designed and used in the experiment, but only one pair that had Ct <30 for all samples was used. Primers for *brp*, forward: AGCT CAAGGACCACATGGACATC; reverse: GCGCCATATCCACCTGGTTGTC (Sigma-Aldrich). Primers for *Ubiquitin-5E*, forward: TCTTCACTTGGTCCTGCGTC; reverse: ATGGCTCGACCTCCAAAGTG (Eurofins Genomics). Primers for *αTubulin84B*, forward: GATCGTGTCCTCGATTACCGC; reverse: GGGA AGTGAATACGTGGGTAGG (Eurofins Genomics).

## Light exposure experiment

Procedures are as previously described (*Sugie et al., 2015*). In brief, animals that hatched within 1 day were transferred to constant darkness (DD, 0 lux) or constant light (LL, 2000–4000 lux by LED panels) environment at 25°C and kept for 2–3 days. Animals were then dissected, and Brp::rGFP cluster number in individual R8 photoreceptor terminals in the medulla was quantified.

## Associative conditioning

For experiments in *Figure 7C and D*, adult flies at 3–7 days after eclosion were kept in a 25°C incubator before and after conditioning. Approximately 50 flies were put in a training tube that is set inside the climate box for 2 min before the conditioning. Temperature in the climate box was set at 24°C, and the relative humidity was set to 70%. For the paired group, 2% 4-methylcyclohexanol (4-MCH, Sigma-Aldrich, diluted in paraffin oil, Sigma-Aldrich) was presented in a cup with the diameter of 5 mm, together with 12 pulses of electric shocks at 90 V for 1 min inside the training tube. The unpaired group received the same 1 min electric shock first, followed by 1 min rest, and then the same 1 min 4-MCH presentation. Flies were dissected at 3 min, 20 min, 90 min, 270 min, and 1 day after conditioning. Experiments were performed multiple times on different days for a single group.

For measuring the memory decay curve of single-odor conditioning, 2% 4-MCH and pure paraffin oil were used for the training. One odor was presented with electric shocks for 1 min, followed by 1 min rest, then another odor was presented without electric shocks. For each reciprocal, preference was tested between 2% 4-MCH and paraffin oil. For the differential odor conditioning and short-term memory performance test, 10% 4-MCH and 10% 3-OCT were used for the conditioning, following the same training protocol. Short-term memory performance was tested immediately after the conditioning. Performance index was calculated as $\#\left(CS-\right)-\#\left(CS+\right)/\#\left(CS-\right)+\#\left(CS+\right)$.

## Sample preparation

Animals used in experiments involving comparisons between control and experimental groups were dissected on the same day. Data were collected from multiple batches of experiments performed on different days. Flies were anesthetized by ice and placed on ice before dissection. The dissection was performed in cold PBS to collect the brain without the ventral nerve cord. The brains were subsequently fixed by 2% paraformaldehyde for 1 hr and then washed 3×20 min by 0.1% PBT (0.1% Triton X-100 in PBS) in a PCR tube, typically 5–6 brains in one tube. Samples were mounted on microscope slides using mounting medium SeeDB2S (*Ke et al., 2016*). For nc82 immunostaining, fixed brains were washed 3×20 min with 0.1% PBT and incubated in 3% normal goat serum (Sigma-Aldrich; G9023) in 0.1% PBT blocking solution for 1 hr at room temperature. The nc82 antibody (DSHB) solution was

diluted 1:20 in the blocking solution. Samples were incubated in antibody solutions at 4°C for 48 hr for both primary and secondary antibodies.

## Confocal imaging

The image acquisition was performed using an Olympus FV1200 confocal microscope platform equipped with both PMT and GaAsP high sensitivity detectors and a 60×/1.42 NA oil immersion objective (PLAPON60XO, Olympus). The Kalman filter was turned on to 2× averaging while scanning. For imaging Brp::rGFP clusters in MB-projecting neurons, scanning speed was set at 2–4 µs/pixel with a voxel size of 0.079×0.079 laterally and 0.370 µm axially. For experimental point spread function (PSF) imaging, SeeDB2S immersed beads were scanned in a setting that is 60×/1.42 NA oil immersion objective; 473 nm laser power: 2.0%; 559 nm laser power: 1.5%; voxel size: 0.079 µm×0.079 µm×0.370 µm; scanning speed: 4 µs/pixel, to produce multiple bead images.

## PSF acquisition and image deconvolution

Sub-resolution fluorescent beads (Tetraspeck Microspheres 0.1 µm, Thermo Fisher Scientific, T7279) were imaged to generate the PSF. The bead solution was diluted with distilled water and sonicated several times to eliminate aggregation. PSF images acquired were processed using Amira software (Thermo Fisher Scientific) with the Extract Point Spread Function module. The PSFs extracted from each image were averaged to create a single PSF for later image deconvolution (resizing voxel size: 0.079 µm×0.079 µm×0.370 µm; 32 pixels×32 pixels×21 slices of image size). Image deconvolution was performed by using the Richardson-Lucy iterative non-blind algorithm in a Fiji plugin DeconvolutionLab2 (*Sage et al., 2017*) or by CLIJ2 GPU-based Richardson-Lucy deconvolution (*Haase et al., 2020*). Images were deconvolved 20 times iteratively to improve image quality.

## Brp clusters segmentation and data analysis

For Brp::rGFP cluster detection, the image stack size was adjusted to double the XY plate pixel number using Fiji to increase the detection precision. The images were subsequently processed using the '3D maxima finder' and the '3D spot segmentation' function of the '3D suite' package in Fiji. 3D maxima were identified for each cluster and used as the starting point for pixel clustering (*Ollion et al., 2013*). An intensity threshold (threshold) was first set to reduce noise. Pixels having values below the threshold would become NaN (not a number). The 'noise tolerance' was then set in the 3D maxima finder to adjust its sensitivity. To be noticed, the threshold is only applied to images processed by the 3D maxima finder to generate the 'peak' image stack. 3D spot segmentation function uses both the 'peak' image and the deconvolved Brp::rGFP image to generate '3D ROIs'. 3D ROIs were then applied on the raw images to quantify the volume (total voxel number in ROI) and Brp::rGFP intensity (total pixel value in ROI) of individual clusters using the '3D ROI manager'. For most images, both the threshold and noise tolerance were set to 200–400, where segmentation was optimal based on the F-score benchmarking. 'Local thresholding method' in 3D spot segmentation was set to 'Gaussian fit', and 'SD value' was set to 2.

The above procedures may include a tiling and stitching step for images and data because of the limited computer power. Typically, an image was tiled into 25 sub-stacks with overlapped margins. After being processed separately, the data acquired from all tiled sub-stacks was stitched back with a deletion of overlapping data points to generate the complete detection result. Nearest neighbors were identified for each particle, and spatial distances in between (NND) were calculated using Python. For each sample, mean NND was calculated separately. For a particular Brp::rGFP cluster, the number of surrounding clusters within $r$ ($r=3 \times$ mean NND) spherically including itself is referred to as the 'AZ density'. The 'local intensity' of a particular Brp::rGFP cluster is the average of Brp::rGFP intensity of all clusters surrounding within $r$ ($r=3 \times$ mean NND) including itself. The plot function of Python was used to plot the color-coded 3D reconstructions. A sample code for analyzing NND, AZ density, and local intensity is provided with a demonstration data sheet (*Supplementary file 1* and analysis code).

## Detection optimization

The 3D detection parameters were optimized by matching the 3D maxima detection results to the ground truths. Ground truths were defined by annotating AZs manually, according to the size,

position, and signal pattern of Brp clusters in the images. The F-score was used to assess detection precision. F-score is the harmonic mean of a system's precision and recall value, calculated by the following formula: 2×[(Precision×Recall)/(Precision+Recall)]. The precision is the number of true positives divided by the total number of identified positives, including incorrect identification. The recall is the number of true positives divided by the total number of true data points defined as the ground truth. F-scores shown in *Figure 1—figure supplement 5* were automatically calculated using a customized Fiji macro that aligned 3D maxima with ground truth. Different combinations of noise threshold and noise tolerance were tested for the test images cropped from raw images.

## Statistical analysis

Statistical analysis was performed using GraphPad Prism version 8, 9, and 10 or Python. For statistical comparisons, the Kruskal-Wallis test with original false discovery rate method of Benjamini and Hochberg correction was applied unless otherwise specified. The desired false discovery rate was set to 0.05. Pearson's correlation coefficient (R) was calculated using Python.

## Acknowledgements

We thank all lab members of Tanimoto Lab at Tohoku University for valuable discussions. We thank Dr. Yuki Tsukada at Keio University for crucial advice on Fiji-based image analysis. We thank the Vienna Drosophila Resource Center and Bloomington Drosophila Stock Center for transgenic flies. Funding: Japan Science and Technology Agency JPMJSP2114 (HW) Tohoku University Research Program 'Frontier Research in Duo' (HT) Ministry of Education, Culture, Sports, Science and Technology 20H00519 (HT) Ministry of Education, Culture, Sports, Science and Technology 22H05481 (HT) Ministry of Education, Culture, Sports, Science and Technology 22KK0106 (HT).

## Additional information

### Competing interests

Hiromu Tanimoto: Reviewing editor, eLife. The other authors declare that no competing interests exist.

### Funding

| Funder | Grant reference number | Author |
|---|---|---|
| Japan Science and Technology Agency | JPMJSP2114 | Hongyang Wu |
| Tohoku University | Frontier Research in Duo | Hiromu Tanimoto |
| Ministry of Education, Culture, Sports, Science and Technology | 20H00519 | Hiromu Tanimoto |
| Ministry of Education, Culture, Sports, Science and Technology | 22H05481 | Hiromu Tanimoto |
| Ministry of Education, Culture, Sports, Science and Technology | 22KK0106 | Hiromu Tanimoto |

The funders had no role in study design, data collection and interpretation, or the decision to submit the work for publication.

### Author contributions

Hongyang Wu, Conceptualization, Data curation, Formal analysis, Funding acquisition, Validation, Investigation, Visualization, Methodology, Writing – original draft, Writing – review and editing; Yoh Maekawa, Formal analysis, Investigation, Methodology, Writing – review and editing; Sayaka Eno, Data curation, Formal analysis, Validation, Investigation, Visualization, Writing – original draft, Writing – review and editing; Shu Kondo, Resources, Methodology, Writing – review and editing; Nobuhiro

Yamagata, Supervision, Methodology; Hiromu Tanimoto, Conceptualization, Supervision, Investigation, Writing – original draft, Project administration, Writing – review and editing

## Author ORCIDs
Hongyang Wu ⓘ https://orcid.org/0000-0001-7889-6941
Yoh Maekawa ⓘ https://orcid.org/0009-0000-0677-3627
Sayaka Eno ⓘ https://orcid.org/0009-0008-0391-5712
Shu Kondo ⓘ https://orcid.org/0000-0002-4625-8379
Nobuhiro Yamagata ⓘ https://orcid.org/0000-0003-1993-2038
Hiromu Tanimoto ⓘ https://orcid.org/0000-0001-5880-6064

Reviewer #1 (Public review): https://doi.org/10.7554/eLife.107663.3.sa1
Reviewer #2 (Public review): https://doi.org/10.7554/eLife.107663.3.sa2
Reviewer #3 (Public review): https://doi.org/10.7554/eLife.107663.3.sa3
Author response https://doi.org/10.7554/eLife.107663.3.sa4

---

# Additional files

## Supplementary files
Supplementary file 1. Example data of a dorsal paired medial (DPM) neuron generated by the 3D spot segmentation, before being processed by the analysis code.

MDAR checklist

Source data 1. Numerical data.

Source code 1. Python code used for nearest neighbor, local intensity, and active zone (AZ) density analysis.

## Data availability
All data generated or analyzed during this study are included in the manuscript and supporting files; source data files have been provided. Further information and requests on resources and reagents should be directed to and will be fulfilled by the corresponding author, Dr. Hiromu Tanimoto (hiromut@m.tohoku.ac.jp).

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
