## [Editor Report · eLife Assessment]

This **important** work introduces a splitGFP-based labeling tool with an analysis pipeline for the synaptic scaffold protein bruchpilot, with tests in the adult Drosophila mushroom bodies, a learning center in the Drosophila brain. The evidence supporting the conclusions is **convincing**.

---

## [Referee Report · Reviewer #1 (Public review)]

Summary:

The study by Wu et al. uses endogenous bruchpilot expression in a cell-type-specific manner to assess synaptic heterogeneity in adult *Drosophila melanogaster* mushroom body output neurons. The authors performed genomic on locus tagging of the presynaptic scaffold protein bruchpilot (brp) with one part of splitGFP (GFP11) using the CRISPR/Cas9 methodology and co-expressed the other part of splitGFP (GFP1-10) using the GAL4/UAS system. Upon expression of both parts of splitGFP, fluorescent GFP is assembled at the C-terminus of brp, exactly where brp is endogenously expressed in active zones. For manageable analysis, a high-throughput pipeline was developed. This analysis evaluated parameters like location of brp clusters, volume of clusters, and cluster intensity as a direct measure of the relative amount of brp expression levels on site using publicly available 3D analysis tools that are integrated in Fiji. Analysis was conducted for different mushroom body cell types in different mushroom body lobes using various specific GAL4 drivers. Further validation was provided by extending analysis to R8 photoreceptors that reside in the fly medulla. To test this new method of synapse assessment, Wu et al. performed an associative learning experiment in which an odor was paired with an aversive stimulus and found that in a specific time frame after conditioning, the new analysis solidly revealed changes in brp levels at specific synapses that are associated with aversive learning. Additionally, brp levels were assessed in R8 photoreceptor terminals upon extended exposure to light.

Strengths:

Expression of splitGFP bound to brp enables intensity analysis of brp expression levels as exactly one GFP molecule is expressed per brp. This is a great tool for synapse assessment. This tool can be widely used for any synapse as long as driver lines are available to co-express the other part of splitGFP in a cell-type-specific manner. As neuropils and thus brp label can be extremely dense, the analysis pipeline developed here is very useful and important. The authors have chosen an exceptionally dense neuropil - the mushroom bodies - for their analysis and compellingly show that brp assessment can be achieved even with such densely packed active zones. The result that brp levels change upon associative learning in an experiment with odor presentation paired with punishment is likewise compelling and strongly suggests that the tool and pipeline developed here can be used in an in vivo context. Thus, the tool and its uses have the potential to fundamentally advance protein analysis not only at the synapse but especially there.

Weaknesses:

The weaknesses I perceived originally were satisfactorily explained and refuted.

---

## [Referee Report · Reviewer #2 (Public review)]

Summary:

The authors developed a cell-type-specific fluorescence-tagging approach using a CRISPR/Cas9 induced spilt-GFP reconstitution system to visualize endogenous Bruchpilot (BRP) clusters at presynaptic active zones (AZ) in specific cell types of the mushroom body (MB) in the adult Drosophila brain. This AZ profiling approach was implemented in a high-throughput quantification process allowing to compare synapse profiles within single cells, cell-types, MB compartments and between different individuals. Aim is to in more detail analyze neuronal connectivity and circuits in this center of associative learning, notoriously difficult to investigate due to the density of cells and structures within the cells. The authors detect and characterize cell-type specific differences in BRP-dependent profiling of presynapses in different compartments of the MB, while intracellular AZ distribution was found to be stereotyped. Next to the descriptive part characterizing various AZ profiles in the MB, the authors apply an associative learning assay and Rab3 knock-down and detected consequent AZ reorganization.

Strengths:

The strength of this study lies in the outstanding resolution of synapse profiling in the extremely dense compartments of the MB. This detailed analysis will serve as an entry point for many future studies of synapse diversity in connection with functional specificity to uncover the molecular mechanisms underlying learning and memory formation and neuronal network logic. Therefore, this approach is of high importance to the scientific community and represents a valuable tool to investigate and correlate AZ architecture and synapse function in the CNS.

Weaknesses:

The results and conclusions presented in this study are conclusively and well supported by the data presented and appropriate controls. As a comment that could possibly aid and strengthen the manuscript (but not required for acceptance of the manuscript): The experiments in the study are based on spilt-GFP lines (BRP:GFP11 and UAS-GFP1-10). The authors clearly validate the new on-locus construct with a genomic GFP insertion (qPCR, confocal and STED imaging of the brain with anti-BRP (Nc82), MB morphology and memory formation). It would be important to comment on the significant overall intensity decrease of anti-BRP (Nc82) in Fig. S1B (R57C10>BRP::rGFP) and possibly a Western Blot with a correlative antibody staining against BRP might help to show that BRP protein level are not affected. Additionally, it would be important to state, at least in the Materials and Methods section, that the flies are not homozygous viable (and to offer an explanation) and to state that all experiments were performed with heterozygous flies.

---

## [Referee Report · Reviewer #3 (Public review)]

Summary:

The authors develop a tool for marking presynaptic active zones in Drosophila brains, dependent on the GAL4 construct used to express a fragment of GFP, which will incorporate with a genome-engineered partial GFP attached to the active zone protein bruchpilot - signal will be specific to the GAL4 expressing neuronal compartment. They then use various GAL4s to examine innervation onto the mushroom bodies to dissect compartment specific differences in the size and intensity of active zones. After a description of these differences, they induce learning in flies with classic odour/electric shock pairing and observe changes after conditioning that are specific to the paired conditioning/learning paradigm.

Strengths:

The imaging and analysis appears strong. The tool is novel and exciting.

Weaknesses:

I feel that the tool could do with a little more characterisation. It is assumed that the puncta observed are AZs with no further definition or characterisation. It is not resolved if the AZs visualised here simply tagged, or are the constructs incorporated to be an active functional part of the AZ.

Comments on revisions:

Apologies, I should have thought of this in the first round of review. An experiment I would suggest (and it is not a difficult one) to address the functionality of the marker: It is mentioned that the genetically tagged half of the construct is homozygous lethal. Can this be placed in trans to a brp null, with a neuronal UAS-expression of the other half of Brp-GFP - Are the animals then (1) alive, and (2) able to fly (brp mutants can't fly, hence the name 'crashpilot') - a rescue would suggest (and that is all that would be needed here) that the reconstituted brp-GFP has function.

On another note, the paper keeps switching between different DAN-GAL4 lines. In 1H, 2Band 4A, there are informative cartoons showing the extension of the neurons for PPL1, APL and DPM neurons - could these be incorporated into figures 5, 6 and 7, and the supplementary figures to help orient the reader. Ideally they would refer to a figure (in Fig 1?) -to refer to the groups of DANs in the adult brain that are known to innervate the MBs (e.g. Fig1 in Mao and Davis, Front in Neural Circuits 2009). I suggest this because I feel that this tool will be widely used, and if non-MB aficionados can follow what's being done here I feel it will be more widely accepted.

---

## [Author Response]

The following is the authors’ response to the original reviews.

**Public Reviews:**

**Reviewer #1 (Public review):**
Summary:The study by Wu et al. uses endogenous bruchpilot expression in a cell-type-specific manner to assess synaptic heterogeneity in adult *Drosophila melanogaster* mushroom body output neurons. The authors performed genomic on locus tagging of the presynaptic scaffold protein bruchpilot (BRP) with one part of splitGFP (GFP11) using the CRISPR/Cas9 methodology and co-expressed the other part of splitGFP (GFP1-10) using the GAL4/UAS system. Upon expression of both parts of splitGFP, fluorescent GFP is assembled at the N-terminus of BRP, exactly where BRP is endogenously expressed in active zones. For manageable analysis, a high-throughput pipeline was developed. This analysis evaluated parameters like location of BRP clusters, volume of clusters, and cluster intensity as a direct measure of the relative amount of BRP expression levels on site, using publicly available 3D analysis tools that are integrated in Fiji. Analysis was conducted for different mushroom body cell types in different mushroom body lobes using various specific GAL4 drivers. To test this new method of synapse assessment, Wu et al. performed an associative learning experiment in which an odor was paired with an aversive stimulus and found that, in a specific time frame after conditioning, the new analysis solidly revealed changes in BRP levels at specific synapses that are associated with aversive learning.Strengths:Expression of splitGFP bound to BRP enables intensity analysis of BRP expression levels as exactly one GFP molecule is expressed per BRP. This is a great tool for synapse assessment. This tool can be widely used for any synapse as long as driver lines are available to co-express the other part of splitGFP in a cell-type-specific manner. As neuropils and thus the BRP label can be extremely dense, the analysis pipeline developed here is very useful and important. The authors have chosen an exceptionally dense neuropil - the mushroom bodies - for their analysis and convincingly show that BRP assessment can be achieved with such densely packed active zones. The result that BRP levels change upon associative learning in an experiment with odor presentation paired with punishment is likewise convincing, and strongly suggests that the tool and pipeline developed here can be used in an in vivo context.Weaknesses:Although BRP is an important scaffold protein and its expression levels were associated with function and plasticity, I am still somewhat reluctant to accept that synapse structure profiling can be inferred from only assessing BRP expression levels and BRP cluster volume. Also, is it guaranteed that synaptic plasticity is not impaired by the large GFP fluorophore? Could the GFP10 construct that is tagged to BRP in all BRP-expressing cells, independent of GAL4, possibly hamper neuronal function? Is it certain that only active zones are labeled? I do see that plastic changes are made visible in this study after an associative learning experiment with BRP intensity and cluster volume as read-out, but I would be reassured by direct measurement of synaptic plasticity with splitGFP directly connected to BRP, maybe at a different synapse that is more accessible.

We appreciate the reviewer’s comments. In the revised manuscript, we have clarified that Brp is an important, but not the only player in the active zone. We have included new data to demonstrate that split-GFP tagging does not severely affect the localization and plasticity of Brp and the function of synapses by showing: (1) nanoscopic localization of Brp::rGFP using STED imaging; (2) colocalization between Brp::rGFP and anti-Brp signals/VGCCs; (3) activity-dependent Brp remodeling in R8 photoreceptors; (4) no defect in memory performance when labeling Brp::rGFP in KCs; These four lines of additional evidence further corroborate our approach to characterize endogenous Brp as a proxy of active zone structure.

**Reviewer #2 (Public review):**
Summary:The authors developed a cell-type specific fluorescence-tagging approach using a CRISPR/Cas9 induced spilt-GFP reconstitution system to visualize endogenous Bruchpilot (BRP) clusters as presynaptic active zones (AZ) in specific cell types of the mushroom body (MB) in the adult Drosophila brain. This AZ profiling approach was implemented in a high-throughput quantification process, allowing for the comparison of synapse profiles within single cells, cell types, MB compartments, and between different individuals. The aim is to analyse in more detail neuronal connectivity and circuits in this centre of associative learning. These are notoriously difficult to investigate due to the density of cells and structures within a cell. The authors detect and characterize cell-type-specific differences in BRP-dependent profiling of presynapses in different compartments of the MB, while intracellular AZ distribution was found to be stereotyped. Next to the descriptive part characterizing various AZ profiles in the MB, the authors apply an associative learning assay and detect consequent AZ re-organisation.Strengths:The strength of this study lies in the outstanding resolution of synapse profiling in the extremely dense compartments of the MB. This detailed analysis will be the entry point for many future analyses of synapse diversity in connection with functional specificity to uncover the molecular mechanisms underlying learning and memory formation and neuronal network logics. Therefore, this approach is of high importance for the scientific community and a valuable tool to investigate and correlate AZ architecture and synapse function in the CNS.Weaknesses:The results and conclusions presented in this study are, in many aspects, well-supported by the data presented. To further support the key findings of the manuscript, additional controls, comments, and possibly broader functional analysis would be helpful. In particular:(1) All experiments in the study are based on spilt-GFP lines (BRP:GFP11 and UAS-GFP1-10).The Materials and Methods section does not contain any cloning strategy (gRNA, primer, PCR/sequencing validation, exact position of tag insertion, etc.) and only refers to a bioRxiv publication. It might be helpful to add a Materials and Methods section (at least for the BRP:GFP11 line). Additionally, as this is an on locus insertion the in BRP-ORF, it needs a general validation of this line, including controls (Western Blot and correlative antibody staining against BRP) showing that overall BRP expression is not compromised due to the GFP insertion and localizes as BRP in wild type flies, that flies are viable, have no defects in locomotion and learning and memory formation and MB morphology is not affected compared to wild type animals.

We thank the reviewer for suggesting these important validations. We included details of the design of the construct and insertion site to the Methods section, performed several new experiments to validate the split-GFP tagging of Brp, and present the data in the revision.

First, to examine whether the transcription of the brp gene is unaffected by the insertion of GFP_11_, we conducted qRT-PCR to compare the *brp* mRNA levels between brp::GFP_11_, *UAS-GFP1-10* and *UAS-GFP1-10* and found no difference (Figure 1 - figure supplement 1A).

To further verify the effect of GFP_11_ tagging at the protein level, we performed anti-Brp (nc82) immunohistochemistry of brains where GFP is reconstituted pan-neuronally. We found unaltered neuropile localization of nc82 signals (Figure 1 - figure supplement 1C). In presynaptic terminals of the mushroom body calyx, we found integration of Brp::rGFP to nc82 accumulation (Figure 1D). We performed super-resolution microscopy to verify the configuration of Brp::rGFP and confirmed the donut-shape arrangement of Brp::rGFP in the terminals of motor neurons (see Wu, Eno et al., 2025 PLOS Biology), corroborating the nanoscopic assembly of Brp::rGFP at active zones (Kittel et al., 2006 Science).

Furthermore, co-expression of RFP-tagged voltage-gated calcium channel alpha subunit Cacophony (Cac) and Brp::rGFP in PAM-γ5 dopaminergic neurons revealed strong presynaptic colocalization of their punctate clusters (Figure 1E), suggesting that rGFP tagging of Brp did not damage key protein assembly at active zones (Kawasaki et al., 2004 J Neuroscience; Kittel et al., Science).

These lines of evidence suggest that the localization of endogenous Brp is barely affected by the C-terminal GFP_11_ insertion or GFP reconstitution therewith. This is in line with a large body of studies confirming that the N-terminal region and coiled-coil domains, but not the C-terminal, region of Brp are necessary and sufficient for active zone localization (Fouquet et al., 2009 J Cell Biol; Oswald et al., 2010 J Cell Biol; Mosca and Luo, 2014 eLife; Kiragasi et al., 2017 Cell Rep; Akbergenova et al., 2018 eLife; Nieratschker et al., 2009 PLoS Genet; Johnson et al., 2009 PLoS Biol; Hallermann et al., 2010 J Neurosci). We nevertheless report homozygous lethality and found the decreased immunoreactive signals in flies carrying the GFP_11_ insertion (Figure 1 - figure supplement 1B).

For these reasons, we always use heterozygotes for all the experiments therefore there is no conspicuous defect in locomotion as reported in the original study (Wagh et al., 2005 Neuron). To functionally validate the heterozygotes, we measured the aversive olfactory memory performance of flies where GFP reconstitution was induced in Kenyon cells using *R13F02-GAL4*. We found that all these transgenes did not alter mushroom body morphology (Figure 7 - figure supplement 1) or memory performance as compared to wild-type flies (Figure 7 - figure supplement 2), suggesting the synapse function required for short-term memory formation is not affected by split-GFP tagging of Brp.

(2) Several aspects of image acquisition and high-throughput quantification data analysis would benefit from a more detailed clarification.(a) For BRP cluster segmentation it is stated in the Materials and Methods state, that intensity threshold and noise tolerance were "set" - this setting has a large effect on the quantification, and it should be specified and setting criteria named and justified (if set manually (how and why) or automatically (to what)). Additionally, if Pyhton was used for "Nearest Neigbor" analysis, the code should be made available within this manuscript; otherwise, it is difficult to judge the quality of this quantification step.(b) To better evaluate the quality of both the imaging analysis and image presentation, it would be important to state, if presented and analysed images are deconvolved and if so, at least one proof of principle example of a comparison of original and deconvoluted file should be shown and quantified to show the impact of deconvolution on the output quality as this is central to this study.

We thank the reviewer for suggesting these clarifications. We have included more description to the revised manuscript to clarify the setting of segmentation, which was manually adjusted to optimize the F-score (previous Figure 1D, now moved to Figure 1 -figure supplement 5). We have included the code used for analyzing nearest neighbor distance, AZ density and local Brp density in the revised manuscript (Supplementary file 1), together with a pre-processed sample data sheet (Supplementary file 2).

Regarding image deconvolution, we have clarified the differential use of deconvolved and not-deconvolved images in the revised manuscript. We have also included a quantitative evaluation of Richardson-Lucy iterative deconvolution (Figure 1 - figure supplement 4). We used 20 iterations due to only marginal FWHM improvement beyond this point (Figure 1 - figure supplement 4).

(3) The major part of this study focuses on the description and comparison of the divergent synapse parameters across cell-types in MB compartments, which is highly relevant and interesting. Yet it would be very interesting to connect this new method with functional aspects of the heterogeneous synapses. This is done in Figure 7 with an associative learning approach, which is, in part, not trivial to follow for the reader and would profit from a more comprehensive analysis.(a) It would be important for the understanding and validation of the learning induced changes, if not (only) a ratio (of AZ density/local intensity) would be presented, but both values on their own, especially to allow a comparison to the quoted, previous AZ remodelling analysis quantifying BRP intensities (ref. 17, 18). It should be elucidated in more detail why only the ratio was presented here.

We thank the reviewer for the suggestion on the presentation of learning-induced Brp remodeling. The reported values in Figure 7C are the correlation coefficient of AZ density and local intensity in each compartment, but not the ratio. These results suggest that subcompartment-sized clusters of AZs with high Brp accumulation (Figure 6) undergo local structural remodeling upon associative learning (Figure 7). For clarity, we have included a schematic of this correlation and an example scatter plot to Figure 6. Unlike the previous studies (refs 17 and 18), we did not observe robust learning-dependent changes in the Brp intensity, possibly due to some confounding factors such as overall expression levels and conditioning protocols as described in the previous and following points, respectively.

(b) The reason why a single instead of a dual odour conditioning was performed could be clarified and discussed (would that have the same effects?).(c) Additionally, "controls" for the unpaired values - that is, in flies receiving neither shock nor odour - it would help to evaluate the unpaired control values in the different MB compartments.

We use single odor conditioning because it is the simplest way to examine the effect of odor-shock association by comparing the paired and unpaired group. Standard differential conditioning with two odors contains unpaired odor presentation (CS-) even in the ‘paired’ group. We now show that single-odor conditioning induces memory that lasts one day as in differential conditioning (Figure 7B; Tully and Quinn, J Comp Phys A 1985).

(d) The temporal resolution of the effect is very interesting (Figure 7D), and at more time points, especially between 90 and 270 min, this might raise interesting results.

The sampling time points after training was chosen based on approximately logarithmic intervals, as the memory decay is roughly exponential (Figure 7B). This transient remodeling is consistent with the previous studies reporting that the Brp plasticity was short-lived (Zhang et al., 2018 Neuron; Turrel et al., 2022 Current Biol).

(e) Additionally, it would be very interesting and rewarding to have at least one additional assay, relating structure and function, e.g. on a molecular level by a correlative analysis of BRP and synaptic vesicles (by staining or co-expression of SV-protein markers) or calcium activity imaging or on a functional level by additional learning assays.

We thank the reviewer for raising this important point. We have performed calcium imaging of KC presynaptic terminals to correlate the structure and function in another study (see Figure 2 in Wu, Eno et al., 2025 PLOS Biology for more detail). The basal presynaptic calcium pattern along the γ compartments is strikingly similar to the compartmental heterogeneity of Brp accumulation (see also Figure 2 in this study). Considering colocalization of other active-zone components, such as Cac (Figure 1E), we propose that the learning-induced remodeling of local Brp clusters should transiently modulate synaptic properties.

As a response to other reviewers’ interest, we used Brp::rGFP to measure different forms of Brp-based structural plasticity upon constant light exposure in the photoreceptors and upon silencing *rab3* in KCs. Since these experiments nicely reproduced the results of previous studies (Sugie et al., Neuron 2013; Graf et al., Neuron 2009), we believe the learning-induced plasticity of Brp clustering in KCs has a transient nature.

**Reviewer #3 (Public review):**
Summary:The authors develop a tool for marking presynaptic active zones in Drosophila brains, dependent on the GAL4 construct used to express a fragment of GFP, which will incorporate with a genome-engineered partial GFP attached to the active zone protein bruchpilot - signal will be specific to the GAL4-expressing neuronal compartment. They then use various GAL4s to examine innervation onto the mushroom bodies to dissect compartment-specific differences in the size and intensity of active zones. After a description of these differences, they induce learning in flies with classic odour/electric shock pairing and observe changes after conditioning that are specific to the paired conditioning/learning paradigm.Strengths:The imaging and analysis appear strong. The tool is novel and exciting.Weaknesses:I feel that the tool could do with a little more characterisation. It is assumed that the puncta observed are AZs with no further definition or characterisation.

We performed additional validation on the tool, including (1) nanoscopic localization of Brp::rGFP using STED imaging; (2) colocalization between Brp::rGFP and anti-Brp signals/VGCCs (Figure 1D-E); (3) activity-dependent active zone remodeling in R8 photoreceptors (Figure 1F). These will be detailed in our point-by-point response below.

**Recommendations for the authors:**

**Reviewer #1 (Recommendations for the authors):**
(1) The authors keep stating, they profile or assess synaptic structure by analyzing BRP localization, cluster volume, and intensity. However, I do not think that BRP cluster volume and intensity warrant an educated statement about presynaptic structure as a whole. I do not challenge the usefulness of BRP cluster analysis for synapse evaluation, but as there are so many more players involved in synaptic function, BRP analysis certainly cannot explain it all. This should at least be discussed.

It is correct that Brp is not the only player in the active zone. We have included more discussion on the specific role of Brp (line 84 to 89) and other synaptic markers (line 250) and edited potentially misunderstanding text.

(2) I do see that changes in BRP expression were observed following associative learning, but is it certain, that synaptic plasticity is generally unaffected by the large GFP fluorophore? BRP is grabbing onto other proteins, both with its C- and N-termini. As the GFP is right before the stop codon, it should be at the N-terminus. How far could BRP function be hampered by this? Is there still enough space for other proteins to interact?

We thank the reviewer for sharing the concerns. We here provided three lines of evidence to demonstrate that the Brp assembly at active zones required for synaptic plasticity is unaffected by split-GFP tagging.

First, we assessed olfactory memory of flies that have Brp::rGFP labeled in Kenyon cells and found the performance comparable to wild-type (Figure 7 - figure supplement 2), suggesting the Brp function required for olfactory memory (Knapek et al., J Neurosci 2011) is unaffected by split-GFP tagging.

Second, we measured Brp remodeling in photoreceptors induced by constant light exposure (LL; Sugie et al., 2015 Neuron). Consistent with the previous study, we found that LL decreased the numbers of Brp::rGFP clusters in R8 terminals in the medulla, as compared to constant dark condition (DD). This result validates the synaptic plasticity involving dynamic Brp rearrangement in the photoreceptors. We have included this result into the revised manuscript (Figure 1F).

To further validate protein interaction of Brp::rGFP, we focused on Rab3, as it was previously shown to control Brp allocation at active zones (Graf et al., 2009 Neuron). To this end, we silenced *rab3* expression in Kenyon cells using RNAi and measured the intensity of Brp::rGFP clusters in γ Kenyon cells. As previously reported in the neuromuscular junction, we found that *rab3* knock-down increased Brp::rGFP accumulation to the active zones, suggesting that Brp::rGFP represents the interaction with Rab3. We have included all the new data to the revised manuscript (Figure 1 - figure supplement 3).

(3) It may well be that not only active-zone-associated BRP is labeled but possibly also BRP molecules elsewhere in the neuron. I would like to see more validation, e.g., the percentage of tagged endogenous BRP associated with other presynaptic proteins.

To answer to what extent Brp::rGFP clusters represent active zones, we double-labelled Brp::rGFP and Cac::tdTomato (Cacophony, the alpha subunit of the voltage-gated calcium channels). We found that 97% of Brp::rGFP clusters showed co-localization with Cac::tdTomato in PAM-γ5 dopamine neurons terminals (Figure 1E), suggesting most Brp::rGFP clusters represent functional AZs.

(4) Z-size is ~200 nm, while x/y pixel size is ~75 nm during acquisition. How far down does the resolution go after deconvolution?

The Z-step was 370 nm and XY pixel size was 79 nm for image acquisition. We performed 20 iterations of Richarson-Lucy deconvolution using an empirical point spread function (PSF). We found that the effect of deconvolution on the full-width at half maximum (FWHM) of Brp::rGFP clusters improves only marginally beyond 20 iterations, when the XY FWHM is around 200 nm and the XZ FWHM is around 450 nm (Figure 1 - figure supplement 4).

(5) Figure Legend 7: What is a "cytoplasm membrane marker"? Does this mean membrane-bound tdTom is sticking into the cytoplasm?

We apologize for the typo and have corrected it to “plasma membrane marker”.

(6) At the end of the introduction: "characterizing multiple structural parameters..." - which were these parameters? I was under the assumption that BRP localization, cluster volume, and intensity were assessed. I do not see how these are structural parameters. Please define what exactly is meant by "structural parameters".

We apologize for the confusion. By "structural parameters”, we indeed referred to the volume, intensity and molecular density of Brp::rGFP clusters. We have revised the sentence to “Characterizing the distinct parameters and localization of Brp::rGFP cluster.”

(7) Next to last sentence of the introduction: "Characterizing multiple structural parameters revealed a significant synaptic heterogeneity within single neurons and AZ distribution stereotypy across individuals." What do the authors mean by "significant synaptic heterogeneity"?

By “synaptic heterogeneity”, we refer to the intracellular variability of active zone cytomatrices reported by Brp clusters. For instance, the intensities of Brp::rGFP clusters within Kenyon cell subtypes were variable among compartments (Figure 2). Intracellular variability of the Brp concentration of individual active zones was higher in DPM and APL neurons than Kenyon cells (Figure 3). These variabilities demonstrate intracellular synaptic heterogeneity. We have revised the sentence to be more specific to the different characters of Brp clusters.

(8) I do not understand the last sentence of the introduction. "These cell-type-specific synapse profiles suggest that AZs are organized at multiple scales, ranging from neighboring synapses to across individuals." What do the authors mean by "ranging from neighboring synapses to across individuals"? Does this mean that even neighboring synapses in the same cell can be different?

We have revised the sentence to “These cell-type-specific synapse profiles suggest that AZs are spatially organized at multiple scales, ranging from interindividual stereotypy to neighboring synapses in the same cells.”

By “neighboring synapses", we refer to the nearest neighbor similarity in Brp levels in some cell-types (Figure 6A-C), and also the sub-compartmental dense AZ clusters with high Brp level in Kenyon cells (Figure 6D-H). By “across individuals”, we refer to the individually conserved active zone distribution patterns in some neurons (Figure 5).

(9) The title talks about cell-type-specific spatial configurations. I do not understand what is meant by "spatial configurations"? Do you mean BRP cluster volume? I think the title is a little misleading.

By “spatial configuration”, we refer to the arrangement of Brp clusters within individual mushroom body neurons. This statement is based on our findings on the intracellular synaptic heterogeneity (see also response to comment #7). We have streamlined the text description in the revised manuscript for clarity.

**Reviewer #2 (Recommendations for the authors):**
(1) For Figure 3A: exemplary two AZs are compared here, a histogram comparing more AZs would aid in making the point that in general, AZ of similar size have different BRP level (intensities) and how much variation exists.

We have included histograms for Brp::rGFP intensity and cluster volumes to Figure 3 in the revised manuscript.

(2) Line 52: "endogenous synapses" is a confusing term; it's probably meant that the protein levels within the synapse are endogenous and not overexpressed.

We apologize for the confusion and have revised the term to “endogenous synaptic proteins.”

(3) It is not clear from the Materials and Methods section, whether and where deconvolved or not-deconvolved images were used for the quantification pipeline. Please comment on this.

We have now revised the Method section to clarify how deconvolved or not-deconvolved images were differently used in the pipeline.

(4) Line 664 (C) not bold.

We have corrected the error.

(5) 725 "Files" should be Flies.

We have corrected the error.

(6) 727 two times "first".

We have corrected the error.

(7) Figure 7. All (A) etc., not bold - there should be consistent annotation.

We want to thank the reviewer for the detailed proof and have corrected all the errors spotted.

**Reviewer #3 (Recommendations for the authors):**
(1) Has there been an expression of the construct in a non-neuronal cell? Astrocyte-like cell? Any glia? As some sort of control for background and activity?

As the reviewer suggested, we verified the neuronal expression specificity of Brp::rGFP. Using *R86E01-GAL4* and *Amon-GAL4*, we compared Brp::rGFP in astrocyte-like glia and neuropeptide-releasing neurons. We found no Brp::rGFP puncta in the neuropils in astrocyte-like glia compared to neurons, suggesting Brp::rGFP is specific to neurons. We have included this new dataset to the revised manuscript (Figure 1 - figure supplement 2).

(2) Similarly, expression of the construct co-expressed with a channelrhodopsin, and induction of a 'learning'-like regime of activity, similarly in a control type of experiment, expression of an inwardly rectifying channel (e.g. Kir2.1) to show that increases in size of the BRP puncta are truly activity dependent? The NMJ may be an optimal neuron to use to see the 'donut' structures of the AZs and their increase with activity. Also, are these truly AZs we are seeing here? Perhaps try co-expressing cacophony-dsRed? If the GFP Puncta are active zones, then they should be surrounded by cacophony.

We would like to clarify that we did not find Brp::rGFP size increase upon learning. Instead, we demonstrated that associative training transiently remodelled sub-compartment-sized AZ “hot spots” in Kenyon cells, indicated by the correlation of local intensity and AZ density (Figure 6-7).

To demonstrate split-GFP tagging does not affect activity-dependent plasticity associated with Brp, we measured Brp remodeling in photoreceptors induced by constant light exposure (LL; Sugie et al., 2015 Neuron). Consistent with the previous study, we found that LL decreased the numbers of Brp::rGFP clusters in R8 terminals in the medulla, as compared to constant dark condition (DD). This result validates the synaptic plasticity involving dynamic Brp rearrangement in the photoreceptors (Figure 1F).

As the reviewer suggested, we performed the STED microscopy for the larval motor neuron and confirmed the donut-shape arrangement of Brp::rGFP (Wu, Eno et al., PLOS Biol 2025).

Also following the reviewer’s suggestion, we double-labelled Brp::rGFP and Cac::tdTomato (Cacophony, the alpha subunit of the voltage-gated calcium channels). We found that 97% Brp::rGFP clusters showed co-localization with Cac::tdTomato in PAM-γ5 dopamine neurons terminals (Figure 1E), suggesting most Brp::rGFP clusters represent functional AZs.

(3) In the introduction: Intro, a sentence about BRP - central organiser of the active zone, so a key regulator of activity.

We have included a few more sentences about the role Brp in the active zones to the revised manuscript.

(4) Figure 1 E, line 650 'cite the resource here'.

We thank the reviewer for pointing out the error and we have corrected it.

(5) Many readers may not be MB aficionados, and to make the data more accessible, perhaps use a cartoon of an MB with the cell bodies of the neurons around the MB expressing the constructs highlighted so that the reader can have a wider idea of the anatomy in relation to the MB.

We appreciate these comments and have appended cartoons of the MB to figures to help readers understand the anatomy.